# Simultaneously Predicting the Pharmacokinetics of CES1-Metabolized Drugs and Their Metabolites Using Physiologically Based Pharmacokinetic Model in Cirrhosis Subjects

**DOI:** 10.3390/pharmaceutics16020234

**Published:** 2024-02-05

**Authors:** Xin Luo, Zexin Zhang, Ruijing Mu, Guangyu Hu, Li Liu, Xiaodong Liu

**Affiliations:** Center of Drug Metabolism and Pharmacokinetics, China Pharmaceutical University, Nanjing 210009, China; 3321010019@stu.cpu.edu.cn (X.L.); 1821010211@stu.cpu.edu.cn (Z.Z.); juiching@stu.cpu.edu.cn (R.M.); 3321010540@stu.cpu.edu.cn (G.H.)

**Keywords:** carboxylesterase 1, liver cirrhosis, physiologically based pharmacokinetic model, prodrugs, pharmacokinetics

## Abstract

Hepatic carboxylesterase 1 (CES1) metabolizes numerous prodrugs into active ingredients or direct-acting drugs into inactive metabolites. We aimed to develop a semi-physiologically based pharmacokinetic (semi-PBPK) model to simultaneously predict the pharmacokinetics of CES1 substrates and their active metabolites in liver cirrhosis (LC) patients. Six prodrugs (enalapril, benazepril, cilazapril, temocapril, perindopril and oseltamivir) and three direct-acting drugs (flumazenil, pethidine and remimazolam) were selected. Parameters such as organ blood flows, plasma-binding protein concentrations, functional liver volume, hepatic enzymatic activity, glomerular filtration rate (GFR) and gastrointestinal transit rate were integrated into the simulation. The pharmacokinetic profiles of these drugs and their active metabolites were simulated for 1000 virtual individuals. The developed semi-PBPK model, after validation in healthy individuals, was extrapolated to LC patients. Most of the observations fell within the 5th and 95th percentiles of simulations from 1000 virtual patients. The estimated AUC and C_max_ were within 0.5–2-fold of the observed values. The sensitivity analysis showed that the decreased plasma exposure of active metabolites due to the decreased CES1 was partly attenuated by the decreased GFR. Conclusion: The developed PBPK model successfully predicted the pharmacokinetics of CES1 substrates and their metabolites in healthy individuals and LC patients, facilitating tailored dosing of CES1 substrates in LC patients.

## 1. Introduction

Liver cirrhosis (LC) is widely prevalent worldwide and results from a variety of causes including obesity, non-alcoholic fatty liver disease, high alcohol consumption, hepatitis B or C infection, autoimmune diseases, cholestatic diseases and iron or copper overload [1,2]. The Child–Pugh score is often used to classify liver cirrhosis into Child–Pugh A (CP-A), Child–Pugh B (CP-B) and Child–Pugh C (CP-C) according to the severity of LC [3,4]. In addition to the impairment of hepatic functions, LC also leads to remarkable alterations in a series of other physiological parameters such as functional liver volume, hepatic arterial blood flow, portal venous blood flow, glomerular filtration rate (GFR), α-acid glycoprotein, albumin content, drug-metabolizing enzymes and transporters. The alterations may directly affect the pharmacokinetics of drugs [5]. For example, Duthaler et al. investigated the effects of LC on the pharmacokinetics of CYP450 cocktail probes with caffeine (CYP1A2), efavirenz (CYP2B6), flurbiprofen (CYP2C9), omeprazole (CYP2C19), metoprolol (CYP2D6) and midazolam (CYP3A). They found that liver cirrhosis increased the plasma exposure of tested probes, the extent of which depended on the type of probe and LC severity. The calculated ratios of the AUC in patients to that in controls (AUCR) of caffeine, efavirenz, flurbiprofen, omeprazole, metoprolol and midazolam in CP-C patients were 6.2, 0.8, 1.4, 10.5, 4.5 and 6.3, respectively. The calculated AUCR values of omeprazole in CP-A, CP-B and CP-C patients were separately 4.8, 6.5 and 10.5. The AUCR values of probes in LC patients were in line with those in the contents of hepatic CYP450s [6]. LC also affects the renal excretion and intestinal absorption of drugs. Furosemide is primarily eliminated through the kidneys. It was reported [7] that clearance (CL) of furosemide significantly decreased from 154 mL/min in control subjects to 91 mL/min in CP-B or CP-C patients, which mainly resulted from decreases in renal clearance (CL_K_). These results indicate that drug dosage adjustments are necessary for LC patients based on the severity of their condition. Thus, regulatory agencies recommend pharmacokinetic studies of drugs in LC patients [8]. However, conducting pharmacokinetic studies in LC patients can be both costly and time-consuming. More importantly, it is difficult to recruit patients, especially patients with CP-C. Physiologically based pharmacokinetic (PBPK) modeling is considered an ideal technique for predicting the pharmacokinetics of drugs in patients with altered physiology. The alterations in physiological parameters, expression of hepatic drug-metabolizing enzymes and transporters under various degrees of severity of LC have been demonstrated. The possibilities for predicting the pharmacokinetics of drugs in LC patients using the PBPK model have been demonstrated [9].

Carboxylesterase1 (CES1) is one of the most abundant drug-metabolizing enzymes in human livers, constituting approximately 1% of the entire liver proteome. CES1 is responsible for 80–95% of total hydrolytic activity in the liver, which mediates the metabolism of a wide range of drugs, pesticides, environmental pollutants and endogenous compounds [10]. CES1-mediated metabolism leads to the biotransformation of a pharmacologically active drug into its inactive metabolite, as exemplified by methylphenidate hydrolysis. CES1 also mediates the activation of some prodrugs. The typical examples are some angiotensin-converting enzyme inhibitors (such as enalapril, cilazapril and temocapril) and neuraminidase inhibitors (oseltamivir). CES1 also hydrolyzes cholesteryl ester in lipid metabolism in human macrophages and hepatocytes, inferring that CES1 could be a potential drug target for the treatment of metabolic diseases, such as diabetes and atherosclerosis [10,11,12,13]. LC has been demonstrated to significantly downregulate expressions of the hepatic CES1 protein; the CES1 contents in CP-B patients and CP-C patients were decreased to 70% and 30% of those of healthy subjects, respectively, and the CES1 enzyme content in CP-A patients was comparable to that of healthy subjects [9]. On this basis, LC can alter the plasma exposure of its substrate drugs such as enalapril and oseltamivir [14,15]. Moreover, it is worth noting that metabolites of most CES1 substrates (such as enalapril and oseltamivir) are mainly eliminated via renal excretion. LC also injures renal functions, leading to decreases in renal clearance of the metabolites, indicating that alterations in the plasma exposure of metabolites by LC are attributed to the integrated effects of the decreases in hepatic CES1 activity and renal clearance.

This study aimed to develop a semi-PBPK model incorporating alterations in hepatic CES1 activity, liver/renal functions, gastrointestinal transit rate and relevant organ blood flow to simultaneously predict the pharmacokinetics of CES1 drugs and their metabolites in LC patients. Clinical pharmacokinetic studies of CES1 drugs were collected from data published on PubMed based on the following criteria. (1) The tested drug must be metabolized primarily by CES1. (2) Pharmacokinetic parameters (such as AUC or plasma drug concentrations) following intravenous (i.v.) and/or oral (p.o.) administration to liver cirrhosis populations must be available. (3) The clinical pharmacokinetic data might come from different reports. Based on these criteria, nine CES1 substrates were included in the simulations. The nine drugs are primarily metabolized by CES1 and include six prodrugs (enalapril, benazepril, cilazapril, perindopril, temocapril and oseltamivir) and three direct-acting drugs (flumazenil, pethidine and remimazolam). Flumazenil and remimazolam are mainly administered by intravenous injection. Pethidine is administrated via intravenous or oral routes. The remaining drugs are administered as oral immediate-release formulations. The predicted results were compared with clinical studies in patients with different statuses of LC. These results will assist in tailoring dosages of CES1 substrates in LC patients.

## 2. Materials and Methods

### 2.1. General Workflow

The workflow for developing a PBPK model (Figure 1) for LC patients. Initially, a semi-PBPK model (Figure 2) was developed for a virtual population of healthy individuals validated using clinical pharmacokinetic studies in healthy subjects. Then, the developed PBPK model was translated to LC patients by replacing the values of system-specific model parameters. Finally, pharmacokinetic predictions were conducted in 1000 virtual patients individuals and compared with clinical pharmacokinetic data from the literature.

### 2.2. Model Development

A semi-PBPK model was developed to simultaneously predict the pharmacokinetics of CES1 substrate drugs and their metabolites in LC patients. The semi-PBPK model consists of the stomach, intestinal wall, intestinal lumen, portal vein, liver, kidney and systemic compartment, which are connected by the blood circulatory system. The elimination of most drugs mainly occurs in the liver and kidneys. Drugs are administrated via the intravenous route or oral route. It is generally accepted that absorption of most orally administered drugs may occur in the small intestine (duodenum, jejunum and ileum). Absorbed amounts of drugs in the stomach, caecum and colon are minor. The effective permeability coefficient (P_eff_) is used to indicate the absorption capacity of a drug [16]. In the simulation, it was assumed that elimination of the tested drugs only occurred in the liver and kidneys and absorption of drugs only occurred in the small intestine.

All available information on anatomical, physiological and ADME parameters of the tested drugs was collected for the initial model construction (Table 1 and Table 2). Coding and solving of the PBPK model were conducted on WinNonlin 8.1 (Pharsight, St. Louis, MO, USA). The specific code and formulas for the model can be found in the Appendix A. After developing the initial model, parts of the plasma concentration curves of drugs from healthy subjects were used to estimate and optimize some parameters. Subsequently, the developed PBPK model was validated using plasma concentration–time curves from the rest of the clinical studies.

### 2.3. PBPK Model Development in LC Patients

The anatomical and physiological parameters in healthy subjects were replaced with those (Table 1) in LC patients. The LC-induced alterations in parameters related to ADME were estimated according to their values in healthy (HT) subjects and the altered physiological parameters.

For CES1-mediated hepatic metabolism,
(1)$${CL}_{int,CI,CES1}={CL}_{int,HT,CES1}\times f_{CES1}\times f_{liver}$$
where CL_int,CI,CES1_ and CL_int,HT,CES1_ represent the values of CES1-mediated intrinsic clearance in the liver of patients and healthy subjects, respectively. f_CES1_ and f_liver_ represent the ratio of CES1 content in patients to that in healthy subjects and liver volume in patients to that in healthy subjects, respectively.

For hepatic elimination of drugs mediated by other routes,
(2)$${CL}_{int,CI,other}={CL}_{int,HT}\times f_{other}\times f_{liver}$$
where CL_int,cirr,other_ and CL_int,heal,other_ represent the values of intrinsic clearance by other routes in the liver of patients and healthy subjects, respectively. f_other_ is the ratio of other targets’ content in patients to that in healthy subjects.

Among the tested drugs, pethidine binds mainly to α1-acid glycoprotein and the rest bind mainly to albumin [88,93,94,95,96,97,98,99] (no data on binding protein for temocapril, so binding to albumin was assumed based on pka < 7.4, acidic). The free fraction of drugs in patient plasma was estimated using Equation (3) [21]:(3)$$f_{u,p,CI}=\frac{1}{1+\frac{(1-f_{u,p,HT})\times P_{prot,CI}}{P_{prot,HT}\times f_{u,p,HT}}}$$
where f_u,p,CI_, f_u,p,HT_, P_prot,CI_ and P_prot,HT_ represent the unbound fraction of the drug in the plasma of patients and healthy subjects and the concentration of drug-bound proteins in the plasma of patients and healthy subjects, respectively.

It was assumed that the free apparent volume of the distribution of the drug is unaltered; the apparent volume of distribution in cirrhosis patients (V_sys,CI_) was derived from the apparent volume of distribution in healthy subjects, i,e.,
(4)$$V_{sys,CI}=\frac{f_{u,p,CI}}{f_{u,p,HT}}\times V_{sys,HT}$$

Liver cirrhosis also impairs renal function and is characterized by a decrease in the glomerular filtration rate (GFR). The renal intrinsic clearance (CL_int,K,CI_) in patients may be estimated using equation [17]:(5)$${CL}_{int,K,CI}={CL}_{int,K,HT}\times {{GFR}_{CI}/GFR}_{HT}$$
where CL_int,k,HT_, GFR_HT_ and GFR_CI_ represent renal intrinsic clearance in healthy subjects and GFR in healthy subjects and patients, respectively.

LC patients are often accompanied by impairment of the intestinal barrier [100]. The Lactulose/Rhamnose ratio is used to assess intestinal permeability [26]. The ratio of cirrhosis patients to healthy subjects was used to correct the absorption rate constant in LC patients:(6)$$P_{eff,CI}=P_{eff,HT}\times {{LR}_{CI}/LR}_{HT}$$
where P_eff,CI_ and P_eff,HT_ are P_eff_ values in LC patients and healthy subjects, respectively. LR_CI_ and LR_HT_ are, respectively, the Lactulose/Rhamnose ratios in LC patients and healthy subjects.

The four virtual populations (normal population, CP-A, CP-B and CP-C patients) were included in the simulations, each of which contained 1000 virtual individuals. For virtual population validation, each virtual individual was generated independently. CL_int_, CL_int,K_, f_u,b_, V_system_, P_eff_, k_a_, K_L:P_, K_G:P_, and K_K:P_ were used to generate virtual individuals. A random individual could be generated by taking random values in the range of 80–120% of the above parameter values. The 5th and 95th percentiles and average values of the simulation derived from 1000 virtual subjects were obtained. Effects of cirrhosis on the plasma exposure of the tested drugs were indexed as AUCR or C_max_R
(7)$$AUCR=\frac{{AUC}_{CI}}{{AUC}_{HT}}$$

Or
(8)$$AUCR=\frac{{CL}_{HT}}{{CL}_{CI}}$$
(9)$$C_{max}R=\frac{C_{max,CI}}{C_{max,HT}}$$
where AUC_CI_, AUC_HT_, CL_CI_, CL_HT_, C_max,CI_ and C_max,HT_ are, respectively, the AUC, CL and C_max_ of the tested drugs in cirrhosis patients and healthy subjects.

### 2.4. Criterion of the Developed PBPK Model

The PBPK model was considered to be successful if the simulated AUC or C_max_ fell within 0.5- to 2-fold of the observed data or the observed data were within the 5th and 95th percentiles of the simulation derived from 1000 virtual subjects [101].

## 3. Results

### 3.1. Drug Data Set

Nine CES1 drugs, including six prodrugs (enalapril, benazepril, cilazapril, perindopril, temocapril and oseltamivir) and three direct-acting drugs (flumazenil, pethidine and remimazolam), were collected from data published on PubMed based on the following criteria. (1) The tested drug must be metabolized primarily by CES1. (2) Pharmacokinetic parameters (such as AUC or plasma drug concentrations) following intravenous (i.v.) and/or oral (p.o.) administration to liver cirrhosis populations must be available. (3) The clinical pharmacokinetic data might come from different reports. The collected pharmacokinetic parameters and drug information on clinical reports are listed in Table 2 and Table 3, respectively.

#### 3.1.1. Enalapril and Enalaprilat

Enalapril, an angiotensin-converting enzyme inhibitor (ACEI), is a prodrug, which is mainly metabolized to the active product enalaprilat via hepatic CES1 [12,102]. Enalaprilat is eliminated primarily through the kidneys [103]. In plasma, enalapril and enalaprilat are mainly bound to albumin, and their free fractions in plasma are 0.55 and 0.5 [33]. Five clinical reports, including two reports involving liver cirrhosis, were selected in the simulations.

**Table 3 pharmaceutics-16-00234-t003:** Clinical information about CES1 substrates in the simulations.

No	Authors	Drug	Dose (mg)	Analytes	Subjects (n)	Ref
1	Ohnishi A et al., 1989	enalapril maleate	10, p.o	enalapril, enalaprilat	Healthy (7)	[14]
		enalapril maleate	10, p.o	enalapril, enalaprilat	CP-C (7)	
2	Todd PA et al., 1986	enalapril maleate	10, p.o	enalapril, enalaprilat	Healthy (12)	[104]
3	Weisser K et al., 1991	enalapril maleate	10, p.o	enalapril, enalaprilat	Healthy (8)	[105]
4	Dickstein K et al., 1987	enalapril maleate	10, p.o	enalapril, enalaprilat	Healthy (10)	[106]
5	Baba T et al., 1990	enalapril maleate	10, p.o	enalapril, enalaprilat	CP-B (7)	[107]
6	Kaiser G et al., 1989	benazepril HCl	10, p.o	benazepril, benazeprilat	Healthy (59)	[108]
7	Schweizer C et al., 1993	benazepril HCl	10, p.o	benazepril, benazeprilat	Healthy (11)	[109]
8	Sioufi A et al., 1994	benazepril HCl	20, p.o	benazepril, benazeprilat	Healthy (24)	[110]
9	Waldmeier F et al., 1991	benazepril HCl	20, p.o	benazepril, benazeprilat	Healthy (4)	[111]
10	Kaiser G et al., 1990	benazepril HCl	20, p.o	benazepril, benazeprilat	CP-B (12)	[112]
11	Macdonald NJ et al., 1993	benazepril HCl	10, p.o	benazeprilat	Healthy (18)	[113]
12	Massarella J et al., 1989	cilazapril	1.0, 2.5, 5, p.o	cilazapril, cilazaprilat	Healthy (24)	[51]
13	Williams PEO et al., 1990	cilazapril	2.5, p.o	cilazapril, cilazaprilat	Healthy (13)	[114]
14	Gross V et al., 1993	cilazapril	1, p.o	cilazapril, cilazaprilat	Healthy (10)	[115]
		cilazapril	1, p.o	cilazapril, cilazaprilat	CP-B (9)	
15	Williams PEO et al., 1989	cilazapril	1, p.o	cilazapril, cilazaprilat	Healthy (12)	[116]
16	Massarella JW et al., 1989	cilazapril	5, p.o	cilazapril, cilazaprilat	Healthy (16)	[117]
17	Francis RJ et al., 1987	cilazapril	1.25, 2.5, 5,10, p.o	cilazaprilat	Healthy (12)	[118]
18	Lecocq B et al., 1990	perindopril ^a^	4, p.o	perindopril, perindoprilat	Healthy (12)	[119]
19	Tsai HH et al., 1989	perindopril ^a^	8, p.o	perindopril, perindoprilat	CP-A (8)	[120]
20	Thiollet M et al., 1992	perindopril ^a^	8, p.o	perindopril, perindoprilat	CP-B (10)	[121]
21	Lees KR et al., 1988	perindopril ^a^	8, p.o	perindoprilat	Healthy (8)	[122]
22	Furuta S et al., 1993	temocapril HCl	1, p.o	temocapril, temocaprilat	Healthy (6)	[123]
		temocapril HCl	1, p.o	temocapril, temocaprilat	CP-C (7)	
23	Abe M et al., 2006	oseltamivir ^b^	75, p.o	oseltamivir, oseltamivir carboxylate	Healthy (7)	[124]
24	Brewster M et al., 2006	oseltamivir ^b^	75, p.o	oseltamivir, oseltamivir carboxylate	Healthy (18)	[125]
25	Jittamala P et al., 2014	oseltamivir ^b^	75, p.o	oseltamivir, oseltamivir carboxylate	Healthy (12)	[126]
		oseltamivir ^b^	150, p.o	oseltamivir, oseltamivir carboxylate	Healthy (12)	
26	Snell P et al., 2005	oseltamivir ^b^	75, p.o	oseltamivir, oseltamivir carboxylate	CP-B (11)	[15]
27	Amrei R et al., 1990	flumazenil	10 mg, i.v.	flumazenil	Healthy (NA)	[127]
28	Breimer LTM et al., 1991	flumazenil	10/10 min, iv	flumazenil	Healthy (7)	[128]
29	Pomier-Layrargues G et al., 1989	flumazenil	2/5 min, iv	flumazenil	CP-B (8)	[129]
		flumazenil	2/5 min, iv	flumazenil	CP-C (8)	
30	Klotz U et al., 1984	flumazenil	2.5, i.v	flumazenil	Healthy (6)	[81]
31	Janssen U,et al., 1989	flumazenil	30, p.o	flumazenil	Healthy (8)	[130]
		flumazenil	2, i.v; 30, p.o	flumazenil	CP-C (8)	
32	Verbeeck RK et al., 1981	pethidine HCl	25, i.v	pethidine	Healthy (6)	[131]
		pethidine HCl	25, p.o	pethidine	Healthy (6)	
33	Mather LE et al., 1975	pethidine HCl	50, i.v	pethidine	Healthy (4)	[132]
34	Kuhnert BR et al., 1980	pethidine HCl	50, i.v	pethidine	Healthy (7)	[133]
35	Guay DR et al., 1984	pethidine HCl	70, i.v	pethidine	Healthy (8)	[134]
36	Guay DR et al., 1985	pethidine HCl	70, i.v	pethidine	Healthy (8)	[135]
37	Pond SM et al., 1981	pethidine HCl	60, iv; 112, po	pethidine	CP-A (5)	[136]
38	Pond SM et al., 1980	pethidine HCl	54.4, iv; 108.8, po	pethidine	CP-B (4)	[137]
39	Mather LE et al., 1976	pethidine HCl	50, iv; 100, po	pethidine	Healthy (4)	[138]
40	Klotz U et al., 1974	pethidine HCl	63.9, i.v	pethidine	Healthy (8)	[139]
		pethidine HCl	53.1, i.v	pethidine	CP-A (10)	
41	Neal EA et al., 1979	pethidine HCl	56, iv; 56, po	pethidine	Healthy (4)	[140]
		pethidine HCl	56, iv; 56, po	pethidine	CP-A (8)	
42	Sheng XY et al., 2020	remimazolam besylate	1.5425, 3.315, i.v	remimazolam	Healthy (3)	[76]
		remimazolam besylate	4.8675, 6.18, i.v	remimazolam	Healthy (7)	
		remimazolam besylate	13.26, 24.6, i.v	remimazolam	Healthy (8)	
		remimazolam besylate	18.3, i.v	remimazolam	Healthy (10)	
43	Stohr T et al., 2021	remimazolam besylate	10.4, i.v	remimazolam	CP-B (8)	[141]
		remimazolam besylate	8.2, i.v	remimazolam	CP-C (3)	

^a^: Perindopril tert-butylamine; ^b^: Oseltamivir phosphate.

#### 3.1.2. Benazepril and Benazeprilat

Benazepril, a prodrug, is metabolized by hepatic CES1 to the active product benazeprilat [12,102], which shows inhibition of angiotensin-converting enzyme (ACE). Benazeprilat is eliminated via renal excretion. Benazepril and benazeprilat are mainly bound to albumin, belonging to drugs with high plasma binding, and their free fractions in plasma are 0.03 and 0.05 [47], respectively. Six clinical reports, including one report involving liver cirrhosis, were selected in the simulations.

#### 3.1.3. Cilazapril and Cilazaprilat

Cilazapril is also metabolized by hepatic CES1 into cilazaprilat [12,102]. Cilazaprilat is mainly eliminated via the kidneys [52]. Cilazapril and cilazaprilat are mainly bound to albumin, belonging to medium plasma-binding drugs, and their free fractions in plasma are 0.70 and 0.76 [50], respectively. Six clinical reports, including one report involving liver cirrhosis, were selected in the simulations.

#### 3.1.4. Perindopril and Perindoprilat

The prodrug perindopril is mainly metabolized by hepatic CES1 to perindoprilat, which shows inhibition of ACE. The bioavailability of perindopril is 66% [64]. Perindopril is primarily converted to perindoprilat in the liver, and other major metabolites of perindopril are perindopril glucuronide and perindopril lactam [142]. Since it is not clear which isoenzyme of UGT metabolizes perindopril to perindopril glucuronide, the change rate of AUC_0-inf_ (0.62) for metoprolol in cirrhosis was used as a variation coefficient of intrinsic clearance for UGT [143]. Perindoprilat is eliminated via renal excretion. Perindopril and perindoprilat are predominantly bound to albumin. Perindopril shows higher plasma binding (percent binding 60%) than perindoprilat (mean percent binding 15%) [72]. 

Four clinical reports, including two reports involving liver cirrhosis, were selected in the simulations. Cirrhosis in perindopril and perindoprilat only have pharmacokinetic parameters and no specific drug concentration–time profile, so only a comparison of parameters was made.

#### 3.1.5. Temocapril and Temocaprilat

Temocapril is also a prodrug and metabolized by hepatic CES1 to temocaprilat. Temocaprilat is eliminated via both bile and the kidneys. The biliary clearance of temocaprilat was about two times the renal clearance [65]. The CL_int,K_ of temocaprilat was calculated to be 949.84 mL/min [64]. Thus, the CL_bile,m_ of temocaprilat was estimated to be 1899.68 mL/min, assuming that the ratio of CL_bile,m_ to CL_int,K_ was 2.0. Biliary excretion of temocaprilat is considered to be mediated by multidrug resistance-associated protein2 (MRP2) [144]. One clinical report involving both liver cirrhosis patients and healthy subjects was selected in the simulations.

#### 3.1.6. Oseltamivir and Oseltamivir Carboxylate

Oseltamivir, a prodrug, is metabolized via hepatic CES1 [12,102] to its active metabolite oseltamivir carboxylate (OC), which has an antiviral effect. About 80% of an orally administered dose of oseltamivir reaches the systemic circulation as the active metabolite. The absolute bioavailability of the active metabolite from orally administered oseltamivir is 75% [145]. About 60 to 70% of an oral oseltamivir dose appears in urine as the active metabolite and less than 5% as oseltamivir. Oseltamivir carboxylate is primarily eliminated via renal excretion, accounting for 93% of intravenous doses [38]. The CL_int,K_ values of both oseltamivir and oseltamivir carboxylate exceed the GFR, indicating that renal elimination occurs via a combination of glomerular filtration and renal tubular secretion. Both oseltamivir and oseltamivir carboxylate are primarily bound to albumin; their bound fractions in plasma were approximately 42% and less than 3% [36]. Four clinical reports, including one report involving liver cirrhosis, were selected in the simulations.

#### 3.1.7. Flumazenil

Flumazenil, a benzodiazepine receptor antagonist, is usually administered by intravenous injection [83]. Flumazenil is inactivated by hepatic CES1 to flumazenil acid and probably by CYP450-catalyzed N-dealkylation to N-demethylated flumazenil [146]. Flumazenil is predominantly bound to serum albumin, and its plasma protein binding is about 40% [85]. Five clinical reports, including two reports involving liver cirrhosis, were selected in the simulations.

#### 3.1.8. Pethidine

Pethidine (meperidine) is a synthetic opioid commonly used for analgesia in humans. Pethidine is metabolized in the body by two different pathways [88,102]. The primary pathway is hepatic CES1 metabolism to pethidinic acid, an inactive metabolite. Another pathway is N-demethylation by CYP2B6 to normeperidine, a nonopioid active metabolite. The oral bioavailability of pethidine varies from 48–56% [147]. Pethidine was predominantly bound to α1-acid glycoprotein. In the simulation for healthy subjects, the free fraction of pethidine in plasma was 0.418 [88]. Ten clinical reports, including four reports involving liver cirrhosis, were selected in the simulations.

#### 3.1.9. Remimazolam

Remimazolam, an ultrashort-acting sedative agent, is metabolized by hepatic CES1 to an inactive carboxy acid metabolite. The plasma protein binding of remimazolam is approximately 92%, predominantly serum albumin [77]. In the clinic, remimazolam is normally administered intravenously. Two clinical reports, including one report involving liver cirrhosis, were selected in the simulations.

### 3.2. Development of PBPK Model and Validation Using Pharmacokinetic Parameters from Healthy Subjects following i.v. or Oral Administrations

Plasma concentration–time profiles of the tested CES1 substrates and their active metabolites following i.v. or oral administration to healthy subjects were simulated using the developed PBPK model and compared with clinical observations. The results showed that most of the observed data of the tested agents fell within the 5th and 95th percentiles of the simulated data (Figure 3 and Appendix A). The corresponding pharmacokinetic parameters AUC, CL and C_max_ were estimated using the mean of the simulated profiles derived from 1000 virtual individuals and compared with clinical observations (Table 4, Table 5, Table 6, Table 7, Table 8, Table 9, Table 10, Table 11 and Table 12). Most of the simulated pharmacokinetic parameter (AUC, CL and C_max_) values for all drugs were also within two-fold of observations (Table 4, Table 5, Table 6, Table 7, Table 8, Table 9, Table 10, Table 11 and Table 12 and Figure 4). All the results demonstrated that the PBPK model was successfully developed.

### 3.3. Prediction of Pharmacokinetic Profiles for CES1 Substrates and Their Active Metabolites following i.v. or Oral Administration to LC Patients Using the Developed PBPK Model

The developed PBPK model, following validation in healthy subjects, was used to predict the pharmacokinetic profiles of the selected CES1 substrates and their active metabolites following intravenous or oral administration to 1000 virtual LC patients (Figure 4), and their pharmacokinetic parameters were estimated using the mean pharmacokinetic profile derived from 1000 simulations (Table 4, Table 5, Table 6, Table 7, Table 8, Table 9, Table 10, Table 11 and Table 12). The results showed that except for oral pethidine, the majority of the drug concentrations in LC patients were well within the 5th and 95th percentiles of pharmacokinetic profiles derived from 1000 virtual LC patients. Most of the estimated pharmacokinetic parameters were also within 0.5–2.0-fold of observations (Figure 4), indicating that the developed PBPK model can predict alterations in pharmacokinetic behaviors of CES1 substrates and their metabolites in LC patients.

Extents of pharmacokinetic parameters under liver cirrhosis, AUCR and C_max_R were also predicted using the estimated pharmacokinetic parameters (Figure 5 and Figure 6). AUC or C_max_ values may come from different clinical reports or different doses, thus, the AUC or C_max_ values were normalized by dose and their mean values were used for estimating the AUCR or C_max_R. The results showed that the vast majority of the ratios of predicted AUCR and C_max_R are close to observed values, with only a few individual values differing significantly, indicating a good prediction. All these show that the PBPK model successfully predicted the pharmacokinetics of drugs in cirrhosis.

### 3.4. Sensitivity Analysis of Model Parameters

The plasma concentration–time curve of enalapril and enalaprilat following oral administration (10 mg) was used as an example for pharmacokinetic sensitivity. Some parameters such as gastrointestinal motility rate (K_t_), intestinal absorption (P_eff_), hepatic arterial blood flow rates (Q_LA_), portal vein blood flow rates (Q_PV_), hepatic CES1 activity (CL_int,L_), kidney blood flow rates (Q_K_), GFR, f_u,b_ and f_u,b,m_ (free fraction of metabolites in blood) may affect the pharmacokinetics of drugs and were selected for sensitivity analysis. According to the variations in the corresponding parameters listed in Table 1, the variations of Q_PV_ and Q_K_ were set to be 1/2-, 1- and 2-fold; Q_LA_ and CL_int,L_ were 1/3-, 1- and 3-fold; variation in GFR was 0.5-, 1- and 1.5-fold; variation in f_u,b_ was 0.7-, 1- and 1.3-fold for enalapril; and f_u,b,m_ was 0.7-, 1- and 1.3-fold for enalaprilat. A report showed that K_t_ values under diabetic status were lower by about 2-fold compared to healthy subjects [148]. Table 1 also showed that K_t_ values under liver cirrhosis were about 1.3-fold those of healthy subjects. Here, variations of K_t_ were set to be 1/2-, 1- and 2-fold. Highly different P_eff_ values of enalapril were reported [30,149,150]. For example, Thoms et al.’s reported P_eff_ value of enalapril was 0.00125 cm/min [149], while the P_eff_ value of enalapril reported by Chaturvedi et al. was 0.0127 cm/min [150]. Thus, variations in the P_eff_ values of enalapril were set to be 1/3-, 1- and 3-fold. The results (Figure 7) show that these tested parameters affect the pharmacokinetics profile of drugs in varying degrees; their contributions to the AUC of enalapril were P_eff_ > CL_int,L_ > K_t_ > f_u,b_ > Q_PV_ > GFR > Q_K_ > Q_LA_ and to that of enalaprilat were P_eff_ > GFR > CL_int,L_ > K_t_ > f_u,b,m_ > Q_PV_ > Q_K_ > Q_LA_. In addition to impairment of liver failure, LC patients were associated with increases in intestinal transit rates, intestinal permeability of drugs, Q_LA_ and f_u,b_ (due to decreases in plasma-binding protein levels) and decreases in GFR, Q_K_, CES1 activity and Q_PV_, although increases in Q_L_ were reported in CP-C patients. The contributions of LC-induced alterations in K_t_, Q_PV_, CL_int,L_, P_eff_, GFR, Q_K_ and f_u,b_ to the plasma concentrations of enalapril and enalaprilat following an oral dose of enalapril maleate (10 mg) administered to CP-C patients and their integrated effects were also simulated. The results showed that decreases in the CL_int,L_ and increases in the P_eff_ of enalapril increased plasma concentrations of enalapril, while the increases in f_u,b_ and K_t_ and decreases in Q_PV_ obviously decreased plasma concentrations of enalapril following an oral dose of enalapril maleate; the net effects were an increase in the plasma concentrations of enalapril. For enalaprilat, increases in P_eff_ and decreases in GFR, Q_K_ and Q_PV_ significantly increased the plasma concentration profiles of enalaprilat, while decreases in CES1 activity and increases in the K_t_ and f_u,b,m_ of enalaprilat significantly decreased plasma concentrations following oral enalapril maleate administration. Their net effects were to decrease plasma concentrations of enalaprilat (Figure 7Q,R). 

## 4. Discussion

Hepatic CES1 mediates the inactivation of direct-acting drugs or the activation of some prodrugs, most of whose active metabolites are mainly eliminated via the kidneys. In addition to hepatic dysfunction, LC is also associated with alterations in organ blood flow, decreases in plasma protein levels, increases in intestinal permeability of drugs and impairment of renal functions, commonly affecting the pharmacokinetics of CES1 substrate drugs and their metabolites. Both the whole-PBPK model and the semi-PBPK model have been widely applied to predict the pharmacokinetics of drugs, but compared with the whole-PBPK model, semi-PBPK model needs fewer parameters without losing key dynamic information [151], which may avoid overparameterization in the whole-PBPK model. Moreover, the semi-PBPK model may avoid some of the parameter estimation difficulties of whole-PBPK models [152]. The main contributions of the study were the successful development of a semi-PBPK model involving intestinal absorption, hepatic metabolism and renal excretion to simultaneously predict the pharmacokinetic profiles of nine CES1 substrates (six prodrugs and three direct-acting drugs) in both healthy subjects and LC patients. Most clinical observations were within the 5th and 95th percentiles of simulations derived from 1000 virtual subjects. Most of the estimated AUC and C_max_ values were also within 0.5–2.0-fold of observations.

The extent of LC-induced alterations in the plasma exposure of CES1 substrates and their metabolites was also assessed using AUCR and C_max_R. It was found that although most of the clinically observed plasma concentrations for the tested agents were within the 5th and 95th percentiles of simulations, poorly predicted AUCR or C_max_R values were found in benazepril, temocaprilat, perindopril and perindoprilat. The predicted AUCR values of flumazenil and pethidine were lower than the clinical observations. Benazepril and temocaprilat belong to highly bound compounds, and their f_u,b_ values were 0.03 and 0.025, respectively. In general, it is difficult to obtain an accurate plasma-binding measurement for highly bound compounds [153]. In addition to CES1, UGTs also mediate perindopril metabolism [142]. The isoenzyme of UGT involved in the metabolism of perindopril has not been identified. In the simulation, it was assumed that LC-induced alterations in the CL_int, UGT_ of perindopril were similar to that of metoprolol [143]. LC patients with different etiologies show different amounts of hepatic CES1. In addition to CES1, other enzymes also mediate the metabolism of flumazenil [146]. Pethidine is co-metabolized by CES1 and CYP2B6 [88,102]. Several reports have demonstrated extensive interindividual variability in the expression of CYP2B6 [154] and CES1 [102]. All of these factors may be reasons leading to the differences between the predicted and the observed AUCR values, which need further investigation.

In general, LC-induced impairments of hepatic CES1 activity increase the plasma exposure of CES1 substrates, but sensitivity analysis revealed that the increases in the plasma concentrations of CES1 substrates in LC patients were only partially attributed to the impairment of hepatic CES1. Increases in the intestinal permeability of drugs were also observed in LC patients, contributing to increased plasma exposures of CES1 substrates. In contrast, LC-induced increases in intestinal transit rate and decreases in plasma-binding proteins and Q_PV_ obviously decreased the plasma exposure of CES1 substrates, which partly attenuated the increases in the plasma exposures of CES1 substrates caused by liver cirrhosis. Metabolites of the tested CES1 substrates are eliminated via the kidneys. The decreases in the plasma exposure of metabolites induced by the impairment of hepatic CES1 activity were also partly attenuated by LC-induced alterations in GFR and Q_K_. Even under some conditions, levels of the metabolites are increased rather than decreased due to impaired renal function. For example, the AUC values of perindoprilat in CP-A and CP-B patients were obviously higher than those in healthy individuals; the observed AUCR values were 2.89 and 1.2, respectively, which were near to predictions (1.98 in CP-A patients and 2.04 in CP-B patients). These findings may partly explain clinical findings that although liver cirrhosis obviously increases the plasma levels of enalapril and perindopril, the magnitude of serum ACE-lowering effects by the two drugs was fairly comparable between LC patients and healthy subjects [14,120,121].

Plasma levels of the direct-acting drugs flumazenil, pethidine and remimazolam following their administration to LC patients were also successfully simulated. The observed AUCR values of remimazolam in LC patients could not be calculated due to a lack of observed pharmacokinetic parameters in LC patients, contrasting our expectation that the AUCR values in CP-B patients and CP-C patients would be 0.76 and 0.61, which may be explained by the fact that the increased plasma concentration by the impairment of hepatic CES1 may be attenuated by increases in hepatic arterial blood flow and increases in f_u,b_ (Appendix A). The above simulations showed that the LC-induced impairments of hepatic CES1 activity may increase plasma levels of CES1 substrates (parent drug) and decrease plasma levels of their metabolites, if dosage adjustments are dependent on characteristics. For example, although LC obviously increases the plasma levels of enalapril and perindopril, the levels of enalaprilat and perindoprilat and the extent of decreases in serum ACE activity were obviously unaltered [14,120,121], indicating that no dosage adjustment of enalapril and perindopril in LC patients is required. The simulated levels of pethidine in the plasma of LC patients were higher than those in healthy subjects, which explained why the results in LC patients were consistent with the clinical observation that LC enhanced the CNS toxicity of pethidine [155], indicated that reduced dosages of pethidine in patients with hepatic insufficiency are needed [156].

However, this study also has some shortcomings. The predictions for healthy subjects were based on “ideal” healthy subjects (body weight assumed to be 70 kg) without considering gender, body weight, race and genetic variance of CES1. Genetic variation in CES1 also affects the pharmacokinetics of CES1 substrates [102]. During the simulation in LC patients, LC patients were considered “ideal” CP-A, CP-B or CP-C patients without considering LC etiology, gender and race. It was reported that the amount of CES1 protein in patients with hepatitis C cirrhosis was approximately 1.47-fold that of patients with alcoholic cirrhosis [157]. Similarly, it was reported that flumazenil might improve memory in patients with alcoholic cirrhosis but not in patients with nonalcoholic cirrhosis [158]. Moreover, the mean absolute CES1 protein expression in female livers was reported to be 17.3% higher than that in male livers [159].

LC patients are often accompanied by impairment of the intestinal barrier and renal function. LC may impair the intestinal barrier and renal function via various mechanisms [100,160]. The most common causes of LC are chronic liver diseases related to alcohol consumption, hepatitis virus infection, obesity and/or usage of drugs. Alcohol and drugs may directly impair the intestinal barrier. LC also leads to microbial alterations, which affect the intestinal epithelial barrier function directly or indirectly. For example, increased endotoxin levels directly downregulate the expression of intestinal tight junctions. Portal hypertension is a severe consequence of cirrhosis, which may lead to ascites, variceal hemorrhage and an impaired intestinal barrier [100]. LC may impair renal functions via activating the renin–angiotensin system, the sympathetic nervous system or nonosmotic hypersecretion of arginine vasopressin. Moreover, the translocation of bacteria and bacterial products from the intestinal lumen to the mesenteric lymph nodes stimulates inflammatory responses, increasing the production of proinflammatory cytokines. Moreover, the increased circulating levels of endotoxin or bacterial DNA also increase serum levels of cytokines, in turn, impairing renal function [160]. 

## 5. Conclusions

The developed PBPK model may successfully be applied simultaneously to predict the pharmacokinetics of CES1 substrate drugs and their active metabolites in healthy subjects and LC patients. The impact of physiological alteration under different degrees of LC on the pharmacokinetic behaviors of drugs may be accurately simulated. The simulated results will help in deciding whether the dosage of CES1 substrates should be adjusted for LC patients.

## Figures and Tables

**Figure 1 pharmaceutics-16-00234-f001:**
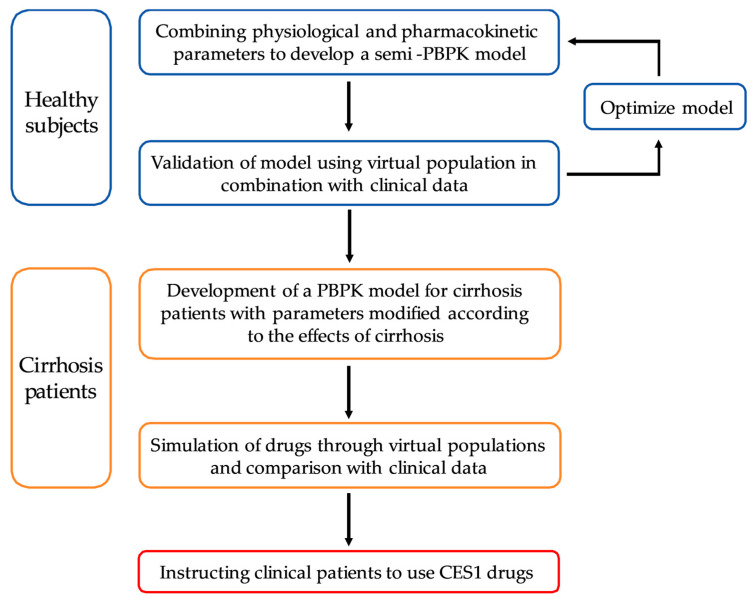
Workflow for developing a semi-PBPK model. It involved establishing a PBPK model in normal subjects and validating it with a virtual population. Afterward, the parameters were changed according to the effects of cirrhosis and a model of PBPK in cirrhosis patients was created. Simulations were performed in virtual populations and compared with clinical pharmacokinetic data.

**Figure 2 pharmaceutics-16-00234-f002:**
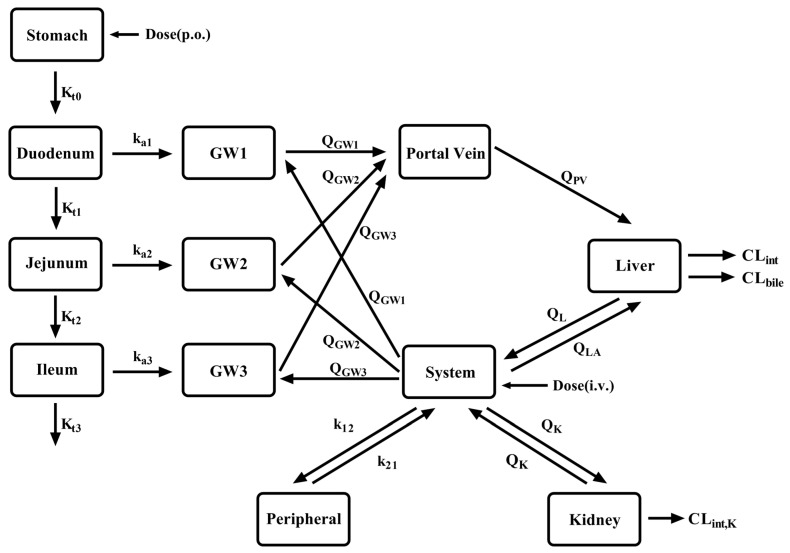
Schematic structure of the semi-PBPK model. K_ti_ represents the gastric emptying rate and intestinal transit rate. GWi represents the gut wall of the duodenum, jejunum and ileum. k_ai_ represents the rate of drug absorption into the gut wall. Q_GWi_ represents the blood flow rate in the gut wall. Q_LA_, Q_L_ and Q_PV_ represent the hepatic artery blood flow rate, hepatic blood flow rate and portal vein blood flow rate, respectively. CL_int_, CL_bile_ and CL_int,K_ represent the intrinsic hepatic clearance, biliary intrinsic clearance and renal intrinsic clearance, respectively.

**Figure 3 pharmaceutics-16-00234-f003:**
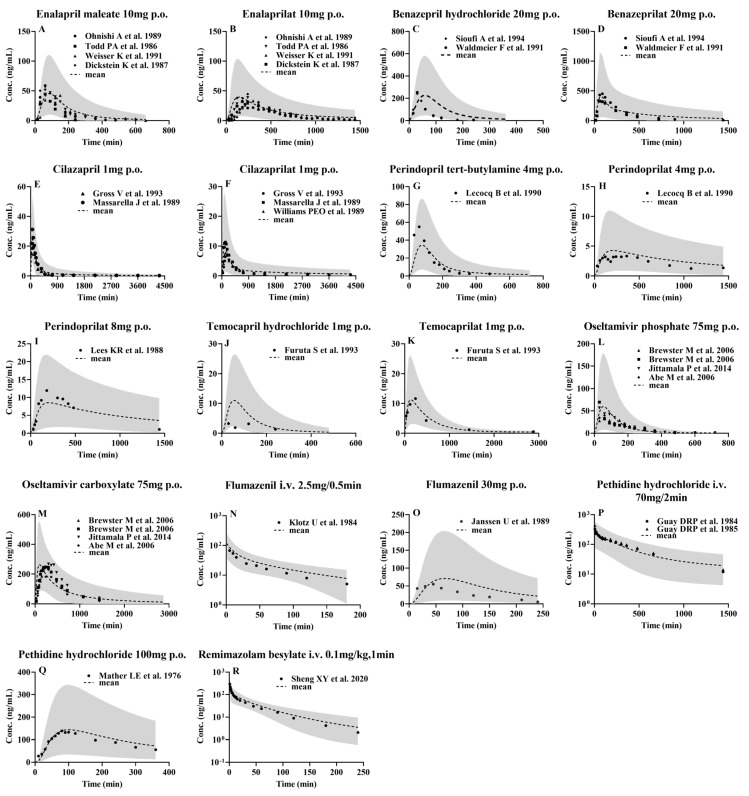
The observed (points) and predicted (lines) plasma concentrations of the tested CES1 substrates and their active metabolites following intravenous or oral administration to healthy subjects. Enalapril (**A**) [14,104,105,106] and enalaprilat (**B**) [14,104,105,106] following oral 10 mg enalapril maleate; benazepril (**C**) [110,111] and benazeprilat (**D**) [110,111] following oral 20 mg benazepril hydrochloride; cilazapril (**E**) [51,115] and cilazaprilat (**F**) [51,115,116] following oral 1 mg cilazapril; perindopril (**G**) [119] and perindoprilat (**H**) [119] following oral 4 mg perindopril tert-butylamine; perindoprilat (**I**) [122] following oral 8 mg perindopril tert-butylamine; temocapril (**J**) [123] and temocaprilat (**K**) [123] following 1 mg temocapril hydrochloride; oseltamivir (**L**) [124,125,126] and oseltamivir carboxylate (**M**) [124,125,126] following oral 75 mg oseltamivir phosphate; flumazenil following intravenous 2.5 mg/0.5 min (**N**) [81] and oral 30 mg (**O**) [130]; pethidine following intravenous 70 mg/2 min pethidine hydrochloride (**P**) [134,135] and oral 100 mg pethidine hydrochloride (**Q**) [138]; remimazolam (**R**) [76] following intravenous 0.1 mg/kg remimazolam besylate. Shaded areas indicate the 5th and 95th percentiles of simulations derived from 1000 virtual individuals. The dashed lines indicate the mean of the simulated profiles.

**Figure 4 pharmaceutics-16-00234-f004:**
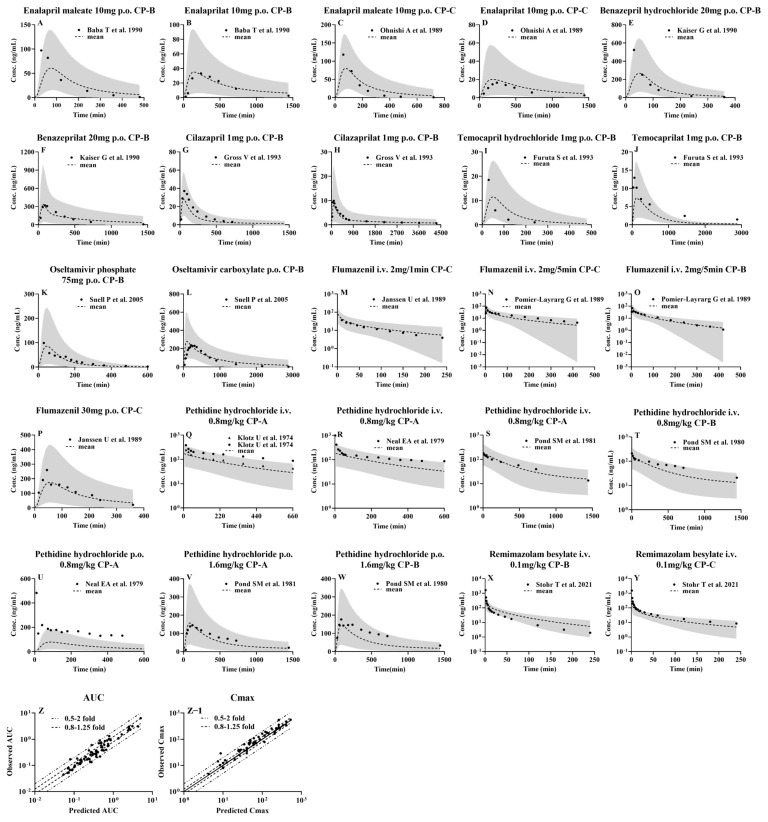
The observed (points) and predicted (lines) plasma concentrations of the tested CES1 substrates and their active metabolites following intravenous or oral administration to LC patients. Enalapril (**A**,**C**) [14,107] and enalaprilat (**B**,**D**) [14,107] following oral 10 mg enalapril maleate to CP-B (**A**,**B**) [107] and CP-C (**C**,**D**) [14]; benazepril (**E**) [112] and benazeprilat (**F**) [112] following oral 20 mg benazepril hydrochloride to CP-B; cilazapril (**G**) [115] and cilazaprilat (**H**) [115] following oral 1 mg cilazapril to CP-B; temocapril (**I**) [123] and temocaprilat (**J**) [123] following oral 1 mg temocapril hydrochloride to CP-B; oseltamivir (**K**) [15] and oseltamivir carboxylate (**L**) [15] following oral 75 mg oseltamivir phosphate to CP-B; flumazenil following intravenous 2 mg/1 min to CP-C (**M**) [130], 2 mg/5 min to CP-C (**N**) [129] and CP-B (**O**) [129]; flumazenil (**P**) [130] following oral 30 mg to CP-C; pethidine following intravenous 0.8 mg/kg,1 min (**Q**) [139], 0.8 mg/kg, 5 min (**R**) [140], 0.8 mg/kg (**S**,**T**) [136,137] pethidine hydrochloride to CP-A (**Q**,**R**,**S**) [136,139,140] and CP-B (**T**) [137]; pethidine following oral 0.8 mg/kg pethidine hydrochloride to CP-A (**U**) [140], 1.6 mg/kg pethidine hydrochloride to CP-A (**V**) [136] and CP-B (**W**) [137]; remimazolam following intravenous 0.1 mg/kg remimazolam besylate to CP-B (**X**) [141] and CP-C (**Y**) [141]. Shaded areas indicate the 5th and 95th percentiles of simulations derived from 1000 virtual individuals. The dashed lines indicate the mean of the simulated profiles. Comparison of the predicted AUC (**Z**) and C_max_ (**Z**−**1**) with observations in healthy subjects and LC patients. Solid, dashed and dotted lines respectively represent unity, 0.8–1.25-fold and 0.5–2-fold errors between observed and predicted data, respectively.

**Figure 5 pharmaceutics-16-00234-f005:**
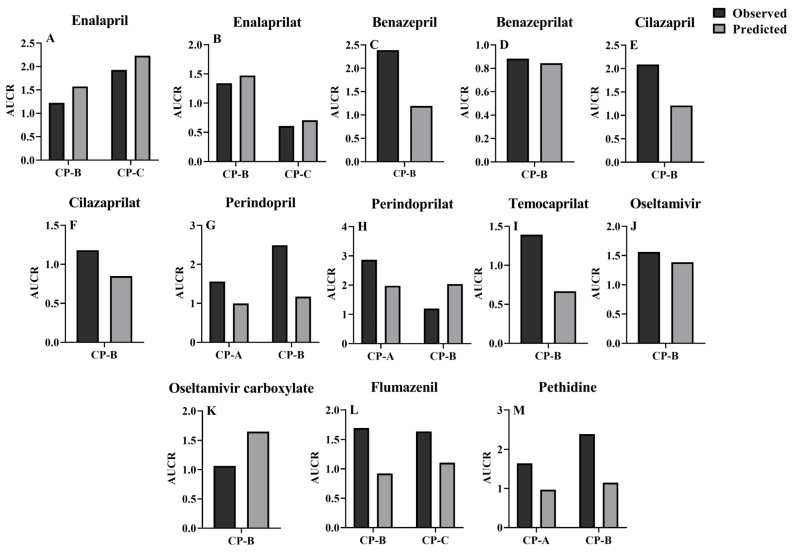
AUCR was calculated from AUC (cirrhotic/healthy) or CL (healthy/cirrhotic) for cirrhotic status and healthy individuals, with the vast majority of parameters in the 0.5–2-fold range. (**A**) Enalapril; (**B**) enalaprilat; (**C**) benazepril; (**D**) benazeprilat; (**E**) I cilazapril; (**F**) cilazaprilat; (**G**) perindopril; (**H**) perindoprilat; (**I**) temocaprilat; (**J**) oseltamivir; (**K**) oseltamivir carboxylate; (**L**) flumazenil; (**M**) pethidine. Parameters not reported in the literature were excluded from the calculations; multiple doses were dose-normalized.

**Figure 6 pharmaceutics-16-00234-f006:**
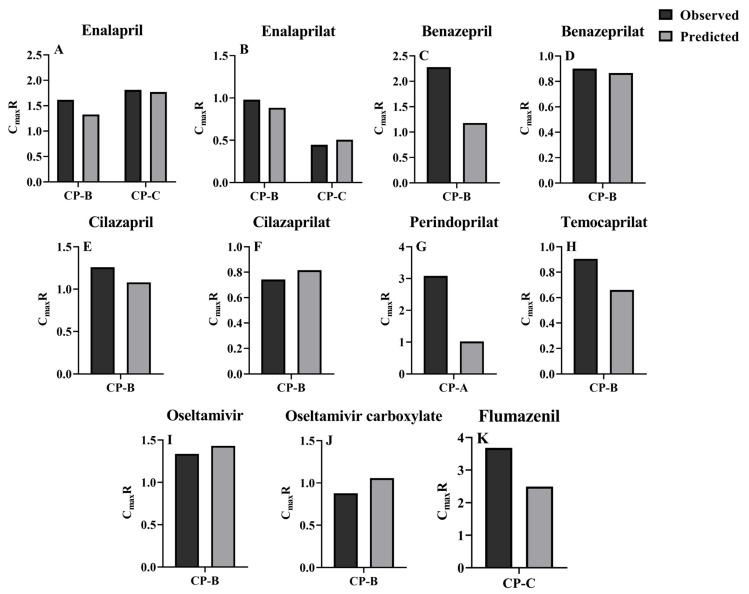
C_max_R was calculated from C_max_ for cirrhotic status and healthy individuals (cirrhotic/healthy), with the vast majority of parameters in the 0.5–2-fold range. (**A**) Enalapril; (**B**) enalaprilat; (**C**) benazepril; (**D**) benazeprilaI (**E**) cilazapril; (**F**) cilazaprilat; (**G**) perindoprilat; (**H**) temocaprilat; (**I**) oseltamivir; (**J**) oseltamivir carboxylate; (**K**) flumazenil. Parameters not reported in the literature were excluded from the calculations; multiple doses were dose-normalized.

**Figure 7 pharmaceutics-16-00234-f007:**
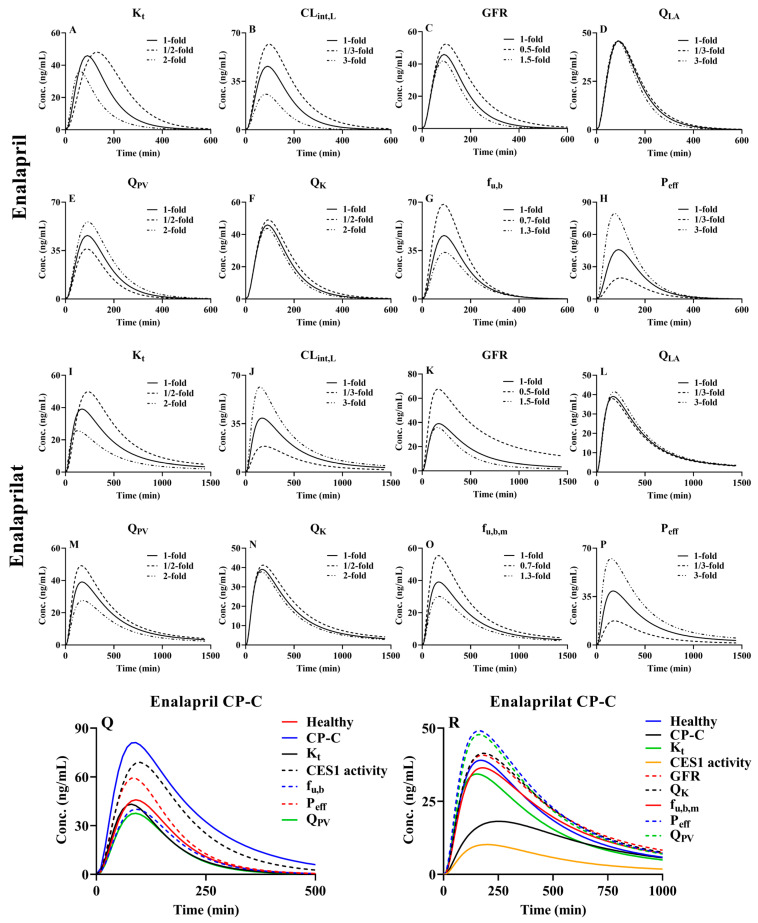
Sensitivity analysis of enalapril and enalaprilat following oral 10 mg enalapril maleate. Enalapril: (**A**) K_t_; (**B**) CL_int,L_; (**C**) GFR; (**D**) Q_LA_; (**E**) Q_PV_; (**F**) Q_K_; (**G**) f_u,b_; (**H**) P_eff_; Enalaprilat: (**I**) K_t_; (**J**) CL_int,L_; (**K**) GFR; (**L**) Q_LA_; (**M**) Q_PV_; (**N**) Q_K_; (**O**) f_u,b,m_; (**P**) P_eff_. f_u,b_ varies by 0.7-fold and 1.3-fold; f_u,b,m_ varies by 0.7-fold and 1.3-fold; GFR varies by 0.5-fold and 1.5-fold; K_t_, Q_PV_ and Q_K_ are varied by 1/2-fold and 2-fold; and the rest are varied by 1/3-fold and 3-fold. Individual contributions of LC-induced alterations in K_t_, CES1 activity, GFR, f_u,b_, P_eff_, Q_K_ and Q_PV_ to plasma concentrations of enalapril (**Q**) and enalaprilat (**R**) following oral 10 mg enalapril maleate administration to LC patients and their integrated effects.

**Table 1 pharmaceutics-16-00234-t001:** Physiological parameters used in the physiologically based pharmacokinetic model in adults with and without cirrhosis.

	Normal	Child–Pugh Class			Units
		A	B	C	
Blood flow rates					
Liver ^a^	1450 [17,18]	1436.5	1176.9	1656.3	mL/min
Hepatic arterial	300 [18]	390 [17,18]	486.9 [9]	1020 [17]	mL/min
Portal vein	1150 [18]	1046.5 [9]	690 [19]	636.3 [9]	mL/min
Kidney	1240 [18]	1091.2 [17]	806 [17]	595.2 [17]	mL/min
Duodenum ^b^	45 [20]	45	45	45	mL/min
Jejunum ^b^	173 [20]	173	173	173	mL/min
Ileum ^b^	102 [20]	102	102	102	mL/min
Volume					
Liver	1690 [18]	1368.9 [21]	1098.5 [21]	895.7 [21]	mL
Portal vein ^b^	70 [18]	70	70	70	mL
Kidney ^b^	280 [18]	280	280	280	mL
Duodenum ^b^	21 [22]	21	21	21	mL
Jejunum ^b^	63 [22]	63	63	63	mL
Ileum ^b^	42 [22]	42	42	42	mL
Transit rates ^c^					
Stomach	0.04 [23]	0.0504 [24]	0.0504 [24]	0.0504 [24]	min^−1^
Duodenum	0.07 [23]	0.0889 [24]	0.0889 [24]	0.0889 [24]	min^−1^
Jejunum	0.03 [23]	0.0381 [24]	0.0381 [24]	0.0381 [24]	min^−1^
Ileum	0.04 [23]	0.0508 [24]	0.0508 [24]	0.0508 [24]	min^−1^
Gut radius					
r1 ^b^	2 [23]	2	2	2	cm
r2 ^b^	1.63 [23]	1.63	1.63	1.63	cm
r3 ^b^	1.45 [23]	1.45	1.45	1.45	cm
Glomerular filtration rate	105 [25]	82 [25]	82 [25]	82 [25]	mL/min
Albumin	44.7 [9]	36.2 [17]	30.4 [17]	26.3 [9]	g/L
α1-acid glycoprotein	0.8 [21]	0.57 [21]	0.52 [21]	0.46 [21]	g/L
CES1	2.45 [9]	2.45 [9]	1.715 [9]	0.735 [9]	mg/g Liver
CYP2B6	17 [21]	17 [21]	15.3 [21]	13.6 [21]	pmol/mg
Lactulose/Rhamnose ratio	0.037 [26]	0.046 [26]	0.052 [26]	0.057 [26]	/
MRP2 ratio	1	0.54 [19]	0.54 [19]	0.54 [19]	/

^a^: Q_L_ = Q_LA_ + Q_PV_, hepatic blood flow rate equals hepatic arterial blood flow rate plus portal vein blood flow rate. ^b^: Assuming that the values are unchanged in cirrhosis. ^c^: Transit rates in cirrhosis were corrected by Table 1 of reference [24].

**Table 2 pharmaceutics-16-00234-t002:** Simultaneously predicting the pharmacokinetics of CES1-metabolized drugs and their metabolites using the physiologically based pharmacokinetic model.

Drug	logP	pka	CL_int_	V_max_	K_m_	K_L;P_ ^d^	K_G;P_ ^d^	K_K;P_ ^d^	CL_b_	V_sys_	K_12_	K_21_	P_eff,A–B_	CL_int,K_	R_b_	f_u,b_	F	k_a_
			mL/min	nmoL/min/ mg protein	μmol/L				mL/min	L	min^−1^	min^−1^	10^−4^ cm/s	mL/min				
Enalapril	0.59 [27]	5.20 [27]	784 [28]	/	/	1.66	2.29	1.79	/	40 [29]	/	/	1.60 [30]	624.6 [31]	0.74 [32]	0.74 [33]	/	
Enalaprilat	−0.74 [33]	2.03 [33]	/	/	/	1.12	1.04	1.25	/	46.1 [34]	0.001 [34]	0.0009 [34]	/	186.4 [35]	0.73 [32]	0.68 [33]	/	
Oseltamivir	0.36 [36]	7.7 [36]	20,255.4 [36]	/	/	1.19	1.12	1.29	/	61.289 [37] ^f^	/	/	/	1357.95 [38]	1 ^e^	0.58 [36]	/	0.061 [39] ^g^
Oseltamivir carboxylate	−1.3 ^a^	4.19 ^a^	/	/	/	1.71	1.89	1.91	/	160.729 [40] ^f^	/	/	/	438.5 [41]	1 ^e^	0.97 [36]	/	
Benazepril	1.11 [42]	4.74 [42]	6696 [43]	/	/	0.087	0.122	0.088	385.8 [44] ^g^	4.8 [45] ^g^	0.0215 [45] ^g^	0.0238 [45] ^g^	1.21 [46]	8391.6 ^c^	1 ^e^	0.03 [47]	0.35 [29]	
Benazeprilat	0.56 [42]	1.97 [42]	/	/	/	0.093	0.088	0.101	/	1.204 [48] ^f^	0.0438 [48] ^f^	0.00837 [48] ^f^	/	447.9 [47]	1 ^e^	0.05 [47]	/	
Cilazapril	0.55 [49]	3.3 [50]	199.7 ^c^	/	/	1.32	1.31	1.43	205 ^a^	18.23 [51] ^f^	0.00325 [51] ^f^	0.00155 [51] ^f^	/	118.095 [52]	1 ^e^	0.7 [49]	/	0.099 [53] ^g^
Cilazaprilat	−0.48 ^a^	3.17 ^a^	/	/	/	1.28	1.22	1.42	/	10.3517 [51] ^f^	0.00084 [51] ^f^	0.008 [51] ^f^	/	75.48 [52]	1 ^e^	0.76 [54]	/	/
Temocapril	2.102 [55]	2.8 [56]	5359.7 [57]	/	/	2.82	3.17	2.47	/	15.398 [58] ^g^	/	/	/	110.2 [59]	1 ^e^	0.3 [60]	0.65 [61]	0.065 [58] ^g^
Temocaprilat	2.215 [62]	2.09 [60]	/	/	/	0.289	0.322	0.251	/	58.535 [63] ^f^	0.00184 [63] ^f^	0.000078 [63] ^f^	/	949.84 [64] 1899.68 [65] ^b^	1 ^e^	0.025 [60]	/	/
Perindopril	−1.31 [66]	3.2 [67]	1011.15 [68] 156.47 [69] ^c,j^	/	/	0.665	0.633	0.742	/	13.119 [70] ^g^	0.0028 [70] ^g^	0.0024 [70] ^g^	1.34 [43]	130.2 [71]	1 ^e^	0.4 [72]	0.66 [64]	/
Perindoprilat	−0.08 ^a^	3.08 ^a^	/	/	/	1.45	1.38	1.61	/	53.44 [73] ^f^	0.271 [73] ^f^	0.0996 [73] ^f^	/	231.78 [74]	1 ^e^	0.85 [72]	/	/
Remimazolam	3.68 ^a^	5.99 ^a^	79,212.96 ^c^	/	/	36.34	63.19	31.2	1180 [75]	15.0768 [76] ^f^	0.01638 [76] ^f^ 0.3117 [76] ^f^ (K_13_)	0.000476 [76] ^f^ 0.5057 [76] ^f^ (K_31_)	/	/	1 ^e^	0.08 [77]	/	
Flumazenil	1.64 [78]	0.86 [79]	8169.9 ^c^	/	/	2.57	2.71	2.41	1120 [80]	24.054 [81] ^g^	0.0376 [81] ^g^	0.0427 [81] ^g^	3.78 [82]	1.67 [83]	1 [84]	0.6 [85]	/	
Pethidine	2.35 [86]	8.7 [86]	/	1.56 [87] ^h^ 5.382 [88] ^i^	261 [87] ^h^ 356 [88] ^i^	14.82	4.18	12.02	/	328.676 [89] ^f^	0.002224 [89] ^f^	0.0003697 [89] ^f^	/	58.78 [90]	0.87 [91]	0.48 [88]	/	0.117 [92] ^g^

^a^: Data from www.drugbank.com, accessed on 4 February 2024; ^b^: Bile intrinsic clearance of temocaprilat; ^c^: Recalculated from CL_L,b_; ^d^: Calculations using Rodgers–Rowland method; ^e^: Assumed values; ^f^: Simulation by WinNonlin, cilazapril and cilazaprilat using 0.5 mg dose pharmacokinetic and remimazolam using 0.025 mg/kg dose pharmacokinetic in simulation; ^g^: Calculated by WinNonlin, flumazenil using T.F. pharmacokinetic to calculate; ^h^: CES1-mediated CL_int_; ^i^: CYP2B6-mediated CL_int_; ^j^: UGT intrinsic clearance of perindopril.

**Table 4 pharmaceutics-16-00234-t004:** Observed and predicted values of AUC_0–t_ and C_max_ of enalapril and enalaprilat following oral enalapril maleate administration to healthy (HT) subjects and liver cirrhosis patients.

Drug	Dose	Subjects	AUC_0–t_ (μg × h/mL)			C_max_ (ng/mL)		
			Obs	Pre	Obs/Pre	Obs	Pre	Obs/Pre
Enalapril	10 mg [14]	HT	0.1229	0.1467	0.84	66.9	45.6	1.47
	10 mg [104]	HT	NR	0.1151	/	NR	45.6	/
	10 mg [105]	HT	0.1600	0.1526	1.05	72.1	45.6	1.58
	10 mg [105]	HT	0.1480	0.1547	0.96	65.4	45.6	1.43
	10 mg [106]	HT	NR	0.1467	/	NR	45.6	/
	10 mg [107]	CP-B	0.1761	0.2253	0.78	110.1	60.6	1.82
	10 mg [14]	CP-C	0.2769	0.3195	0.87	123.4	80.7	1.53
Enalaprilat	10 mg [14]	HT	0.3754	0.3683	1.02	46.1	39.7	1.16
	10 mg [104]	HT	NR	0.3683	/	NR	39.7	/
	10 mg [105]	HT	0.2170	0.3683	0.59	29.3	39.7	0.74
	10 mg [105]	HT	0.2600	0.3683	0.71	37.3	39.7	0.94
	10 mg [106]	HT	NR	0.2776	/	NR	39.7	/
	10 mg [107]	CP-B	0.3812	0.5154	0.74	36.8	35.1	1.05
	10 mg [14]	CP-C	0.1733	0.2476	0.70	16.8	20.1	0.84

NR: Not reported.

**Table 5 pharmaceutics-16-00234-t005:** Observed and predicted values of AUC_0–t_ and C_max_ of benazepril and benazepril following benazepril hydrochloric administration to healthy (HT) subjects and cirrhosis.

Drug	Dose	Subjects	AUC_0–t_ (μg × h/mL)			C_max_ (ng/mL)		
			Obs	Pre	Obs/Pre	Obs	Pre	Obs/Pre
Benazepril	10 mg [108]	HT	0.1390	0.2571	0.54	139.139	113.545	1.23
	10 mg [109]	HT	0.1380	0.2665	0.52	78.957	113.545	0.70
	20 mg [110]	HT	0.2195	0.4611	0.48	265.313	227.089	1.17
	20 mg [111]	HT	NR	0.4611	/	252.98	227.089	1.11
	20 mg [112]	CP-B	0.6159	0.5883	1.05	543.472	268.130	2.03
Benazeprilat	10 mg [113]	HT	1.5330	1.6492	0.93	188.704	198.97	0.95
	10 mg [108]	HT	1.0787	1.3554	0.80	200.410	198.97	1.01
	10 mg [109]	HT	1.1039	1.3554	0.81	164.520	198.97	0.83
	20 mg [110]	HT	2.3800	2.7107	0.88	463.830	397.95	1.17
	20 mg [111]	HT	NR	2.7107	/	342.164	397.95	0.86
	20 mg [112]	CP-B	2.1650	2.3870	0.91	345.010	344.85	1.00

NR: Not reported.

**Table 6 pharmaceutics-16-00234-t006:** Observed and predicted values of AUC_0–t_ and C_max_ of cilazapril following oral cilazapril to healthy (HT) subjects and LC patients.

Drug	Dose	Subjects	AUC_0–t_ (μg × h/mL)			C_max_ (ng/mL)		
			Obs	Pre	Obs/Pre	Obs	Pre	Obs/Pre
Cilazapril	1 mg [51]	HT	0.0998	0.1044	0.96	33.9	26.2	1.29
	2.5 mg [51]	HT	0.2560	0.2610	0.98	82.7	65.4	1.26
	5 mg [51]	HT	0.4960	0.5221	0.95	182.0	130.8	1.39
	2.5 mg [114]	HT	0.1830	0.2341	0.78	75.7	65.4	1.16
	1 mg [115]	HT	0.0657	0.0890	0.74	25.2	26.2	0.96
	1 mg [115]	CP-B	0.1840	0.1201	1.53	40.0	28.3	1.41
Cilazaprilat	1 mg [51]	HT	0.0791	0.0725	1.09	12.4	10.2	1.22
	1 mg [116]	HT	NR	0.1158	/	8.3	10.2	0.81
	2.5 mg [51]	HT	0.175	0.1811	0.97	37.7	25.4	1.48
	5 mg [51]	HT	0.342	0.3623	0.94	94.2	50.8	1.85
	2.5 mg [114]	HT	0.178	0.1811	0.98	39.3	25.4	1.55
	5 mg [117]	HT	0.398	0.6580	0.60	83.4	50.8	1.64
	1.25 mg [118]	HT	0.070	0.0906	0.77	13.0	12.7	1.02
	2.5 mg [118]	HT	0.170	0.1811	0.94	36.0	25.4	1.42
	5 mg [118]	HT	0.280	0.3623	0.77	74.0	50.8	1.46
	10 mg [118]	HT	0.550	0.7246	0.76	165.0	101.5	1.63
	1 mg [115]	HT	0.053	0.0725	0.73	7.96	10.2	0.78
	1 mg [115]	CP-B	0.0775	0.0695	1.12	10.2	8.3	1.23

NR: Not reported.

**Table 7 pharmaceutics-16-00234-t007:** Observed and predicted values of AUC_0_**_–_**_t_ and C_max_ of perindopril following oral perindopril tert-butylamine administration to healthy (HT) subjects and LC patients.

Drug	Dose	Subjects	AUC_0–t_ (μg × h/mL)			C_max_ (ng/mL)		
			Obs	Pre	Obs/Pre	Obs	Pre	Obs/Pre
Perindopril	4 mg [119]	HT	0.121	0.120	1.01	64.2	34.6	1.86
	8 mg [120]	CP-A	0.377	0.239	1.58	NR	70.4	/
	8 mg [121]	CP-B	0.602	0.281	2.14	NR	77.0	/
Perindoprilat	4 mg [119]	HT	0.0520	0.0681	0.76	4.7	4.3	1.09
	8 mg [122]	HT	0.1197	0.1362	0.88	NR	8.5	/
	8 mg [120]	CP-A	0.3210	0.2695	1.19	29	8.8	3.33
	8 mg [121]	CP-B	0.1340	0.2777	0.48	NR	8.6	/

NR: Not reported.

**Table 8 pharmaceutics-16-00234-t008:** Observed and predicted values of AUC_0–t_ and C_max_ of temocapril and temocaprilat following oral temocapril hydrochloride administration to healthy (HT) subjects and LC patients.

Drug	Dose	Subjects	AUC_0–t_ (μg × h/mL)			C_max_ (ng/mL)		
			Obs	Pre	Obs/Ore	Obs	Pre	Obs/Ore
Temocapril	1 mg [123]	HT	NR	0.0257	/	NR	11.0	/
	1 mg [123]	CP-B	NR	0.0271	/	NR	11.6	/
Temocaprilat	1 mg [123]	HT	0.1230	0.1199	1.03	15.8	11.2	1.41
	1 mg [123]	CP-B	0.1714	0.0800	2.14	14.3	7.4	1.93

NR: Not reported.

**Table 9 pharmaceutics-16-00234-t009:** Observed and predicted values of AUC_0_**_–_**_t_ and C_max_ of oseltamivir and oseltamivir carboxylate (OC) following oral oseltamivir phosphate administration to healthy (HT) subjects and cirrhosis.

Drug	Dose	Subjects	AUC_0–t_ (μg × h/mL)			C_max_ (ng/mL)		
			Obs	Pre	Obs/Pre	Obs	Pre	Obs/Pre
Oseltamivir	75 mg [126]	HT	0.1590	0.1430	1.11	74.4	59.8	1.24
	75 mg [125]	HT	0.1240	0.1430	0.87	75.1	59.8	1.26
	75 mg [125]	HT	0.1140	0.1430	0.80	67.6	59.8	1.13
	75 mg [124]	HT	0.1188	0.1442	0.82	61.0	59.8	1.02
	150 mg [126]	HT	0.3130	0.2860	1.09	192.0	119.5	1.61
	75 mg [15]	CP-B	0.2100	0.1985	1.06	100.0	85.6	1.17
Oseltamivir carboxylate	75 mg [126]	HT	3.0200	2.5068	1.20	291.00	264.93	1.10
	75 mg [125]	HT	2.6500	2.5068	1.06	276.00	264.93	1.04
	75 mg [125]	HT	2.5600	2.5068	1.02	278.00	264.93	1.05
	75 mg [124]	HT	3.1763	3.0861	1.03	360.31	264.93	1.36
	150 mg [126]	HT	6.3100	5.0135	1.26	550.00	529.86	1.04
	75 mg [15]	CP-B	3.1000	4.3235	0.72	260.00	279.86	0.93

**Table 10 pharmaceutics-16-00234-t010:** Observed and predicted values of AUC_0–t_ (μg × h/mL) or CL (L/min) and C_max_ (ng/mL) of flumazenil to healthy (HT) subjects and LC patients.

Dose	Subjects	AUC_0–t_ or CL			C_max_		
		Obs	Pre	Obs/Pre	Obs	Pre	Obs/Pre
10 mg, 1 min, i.v. [127]	HT	0.9000 ^a^	0.4486 ^a^	2.01			
10 mg, 10 min, i.v. [128]	HT	0.8967 ^a^	0.4549 ^a^	1.97			
2.5 mg, 0.5 min, i.v. [81]	HT	0.7160 ^a^	0.4766 ^a^	1.50			
2 mg, 5 min, i.v. [129]	CP-B	0.4932 ^a^	0.4988 ^a^	0.99			
2 mg, 5 min, i.v. [129]	CP-C	0.3165 ^a^	0.4030 ^a^	0.79			
2 mg, 1 min, i.v. [130]	CP-C	0.7050 ^a^	0.4295 ^a^	1.64			
30 mg, p.o. [130]	HT	NR	0.1741 ^b^	/	70.1	71.0	0.99
30 mg, p.o. [130]	CP-C	NR	0.5139 ^b^	/	258.0	174.7	1.48

^a^: represent CL; ^b^: represent AUC; NR: Not reported.

**Table 11 pharmaceutics-16-00234-t011:** Observed and predicted values of AUC_0–t_ (μg × h/mL) or CL (L/min) and C_max_ (ng/mL) of pethidine following oral and intravenous pethidine HCl administration to healthy (HT) subjects and LC patients.

Dose	Subjects	AUC_0–t_ or CL			C_max_		
		Obs	Pre	Obs/Pre	Obs	Pre	Obs/Pre
25 mg, 1 min, i.v. [131]	HT	0.5624 ^a^	0.4956 ^a^	1.13			
50 mg, 1 min, i.v. [132]	HT	1.0200 ^a^	0.7784 ^a^	1.31			
50 mg, 1 min, i.v. [133]	HT	0.9640 ^a^	0.8532 ^a^	1.13			
70 mg, 2 min, i.v. [134]	HT	0.7505 ^a^	0.4952 ^a^	1.52			
70 mg, 2 min, i.v. [135]	HT	0.7226 ^a^	0.4952 ^a^	1.46			
0.8 mg/kg, 1 min, i.v. [139]	HT	1.3160 ^a^	0.7887 ^a^	1.67			
0.8 mg/kg, 5 min, i.v. [140]	HT	0.9000 ^a^	0.6972 ^a^	1.29			
0.8 mg/kg, 1 min, i.v. [136]	CP-A	0.3920 ^a^	0.5349 ^a^	0.73			
0.8 mg/kg, 1 min, i.v. [139]	CP-A	0.6640 ^a^	0.7405 ^a^	0.90			
0.8 mg/kg, 5 min, i.v. [140]	CP-A	0.5730 ^a^	0.7560 ^a^	0.76			
0.8 mg/kg, 1 min, i.v. [137]	CP-B	0.3730 ^a^	0.5724 ^a^	0.65			
25 mg, p.o. [131]	HT	0.9270 ^a^	0.8563 ^a^	1.08	NR	36.0	/
100 mg, p.o. [138]	HT	0.8600 ^b^	0.6097 ^b^	1.41	170.0	143.9	1.18
0.8 mg/kg, p.o. [140]	HT	NR	0.4649 ^b^	/	NR	80.6	/
1.6 mg/kg, p.o. [136]	CP-A	NR	1.2681 ^b^	/	NR	157.0	/
0.8 mg/kg, p.o. [140]	CP-A	NR	0.4439 ^b^	/	NR	78.5	/
1.6 mg/kg, p.o. [137]	CP-B	NR	1.1983 ^b^	/	NR	146.3	/

^a^: represent CL; ^b^: represent AUC; NR: Not reported.

**Table 12 pharmaceutics-16-00234-t012:** Observed and predicted values of AUC_0–t_ of remimazolam following intravenous remimazolam besylate administration to healthy (HT) subjects and LC patients.

Dose	Subjects	AUC_0–t_ (μg × h/mL)		
		Obs	Pre	Obs/Pre
0.05 mg/kg [76]	HT	0.0447	0.0536	0.83
0.075 mg/kg	HT	0.0665	0.0787	0.84
0.1 mg/kg	HT	0.0860	0.1000	0.86
0.2 mg/kg	HT	0.1683	0.2145	0.78
0.3 mg/kg	HT	0.2517	0.2960	0.85
0.4 mg/kg	HT	0.3317	0.3979	0.83
10.4 mg [141]	CP-B	NR	0.1277	/
8.2 mg	CP-C	NR	0.0805	/

NR: Not reported.

## Data Availability

Data are contained within the article and the Appendix A.

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
