# Peer review of "Simultaneously Predicting the Pharmacokinetics of CES1-Metabolized Drugs and Their Metabolites Using Physiologically Based Pharmacokinetic Model in Cirrhosis Subjects"

_pharmaceutics, 2024, doi:10.3390/pharmaceutics16020234_

Round 1

Reviewer 1 Report

Comments and Suggestions for Authors

The authors developed a semi-PBPK model to predict 9 CES1 substrates in LC patients. Some issues should be addressed before publication.

Methods:

·         Characteristics of 100 virtual individuals should be specified.

·         Criteria for varying parameters when conducting sensitivity analysis should be justified.

Results:

·         The details of PK of the studied drugs should be summarized in a table to easily capture and compare differences in PK among studied drugs.  

Discussion

·         The clinical applicability of the developed PBPK models is still not clear.

·         The mechanism by which LC injures renal function and increases in intestinal permeability should be discussed. 

Reviewer 2 Report

Comments and Suggestions for Authors

The authors have also reviewed previously conducted studies and as a result there are a lot of references.  There is room for improving the article.

GW stands for in figure 1?

Why CYP2B6 is added in table 1?

Table 2 can be presented in combine form by presenting it in rotated form (this is a suggestion not a compulsion)

The criterion in section 2.4 is authors own or adopted from somewhere? If adopted from elsewhere please give reference.

In result section 3.1 either give clear headings of drug or simply use separate paragraph, e.g. give heading 3.1.1 to Enalapril and enalaprilat (please adopt journals’ format)

In table 3, HCl is written wrongly, in few places it is written as HC1 (digit 1 is there) while is the same table it is written as HCL, please present it appropriately.  

Why the predicted pharmacokinetic parameters (values) are not presented in table form?

Section 3.1 is a generalized section which can be placed in introduction. It contains no results.

The conclusion section is not well elaborated. It must include the conclusion of present study.

Uniform style for references is not used. Please use the specified style.

Reference 16 and 18, it is self-citation and unnecessarily used.  

Comments on the Quality of English Language

The English language need to be improved. 

Reviewer 3 Report

Comments and Suggestions for Authors

For authors:

This study presents equations for a semi-mechanistic PBPK modeling method to predict PK for drugs in hepatically impaired patients. This paper focuses on drugs that are primarily metabolized (or prodrugs that are converted to the active species) by carboxylesterase 1, which is primarily expressed in the liver. Effects of hepatic impairment (eg, differences in organ blood flows, plasma protein binding, functional liver volume, hepatic enzymatic activity, glomerular filtration rate and gastrointestinal transit rate) are incorporated in the model. PBPK models for 6 prodrugs and 3 directly-acting drugs were developed for healthy subjects, and then after the PBPK models were adjusted for the physiological parameters to address the effects of hepatic impairment, simulations for hepatically impaired subjects were conducted and compared to clinical data.

Major comments:

- A figure laying out the workflow/approach would be helpful. For example, the steps to include would be developing semi-mechanistic PBPK models for healthy subjects, refining the models where necessary, incorporating physiological changes with hepatic impairment as appropriate to match the clinical study that was available (eg, CP-B for cilazapril), running the clinical trial simulation, and then comparing output to the observed data.

- The manuscript had a lot of equations, but it might be more effective for the bulk of the equations to go into an appendix. The equations in section 2.3 on are needed, but the equations in section 2.2 could go in an appendix. Regarding Section 2.2, it would be more interesting to the reader to give a high-level description of why the given approach was used (semi-mechanistic instead of full PBPK model) and any key assumptions. For example, the GI tract (absorption) part of the model has only 4 compartments. What is the assumption here, that the drugs included in this analysis were well absorbed (ie, a simple model that does not include the large intestines was acceptable for the current work)?

- For the methods, please include details on how the virtual populations were set up. Were the age range and percentage of male and female set to match the clinical studies? Which parameters changed? What distributions were assumed? Was there a reason 100 subjects were simulated instead of a higher number?

Minor comments:

- There are many places where a space is needed between a word and “(“ but is missing. Examples can be found in the third and fourth line of the abstract, lines 33, 34, 40, 41, 71 and 83.

- The use of “respectively” is sometimes a little confusing. Examples can be found in lines 132 and 167.

- Regarding line 53, “Thus, regulatory agencies have recommended pharmacokinetic studies of drugs in LC patients[8].” Reference 8 does not seem like a good reference for this point. The FDA (Pharmacokinetics in Patients with Impaired Hepatic Function: Study Design, Data Analysis, and Impact on Dosing and Labeling) and/or EMA guidance would be a better reference.

- Regarding line 90, Figure 1 is a schematic diagram, not a workflow.

- Regarding lines 221-223, I did not understand this.

- Line 224 refers to “One hundred virtual populations” but I think this is just a typo. I think what is meant is “One hundred simulated subjects from four virtual populations, ie, for healthy subjects and CP-A, CP-B and CP-C patients…”

- Line 141, deprived should be derived.

- The references for Table 1 parameters do not seem clear. For example, I did not see any of the physiological parameters in reference 18 (mentioned in the table title), the GFR did not match the Morris and Davies paper, and it was not clear where the transit rates were from or how they were calculated. 

- In Table 2, it would be clearer to type out the name for OC instead of use the abbreviation (as was done for other metabolites).

Comments on the Quality of English Language

Minor editing of English language required

Round 2

Reviewer 2 Report

Comments and Suggestions for Authors

The manuscript is improved but still there is room for improvement. Previoulsy to used HCl appropriately. In one place "1" was in place of alphabet "l". Please use standard format i.e. HCl. In table 1, most values have references but some are without references. From where the values without references are adopted.  

Comments on the Quality of English Language

Minor changes required
